# Neurosurgical Treatment of Pain

**DOI:** 10.3390/brainsci12111584

**Published:** 2022-11-20

**Authors:** Rafael G. Sola, Paloma Pulido

**Affiliations:** 1Innovation in Neurosurgery, Department of Surgery, Autonomous University of Madrid, 28049 Madrid, Spain; 2Department of Surgery, Autonomous University of Madrid, 28049 Madrid, Spain

**Keywords:** deep brain stimulation (DBS), dorsal root entry zone (DREZ), motor cortex stimulation (MCS), peripheral nerve stimulation (PNS), spinal cord stimulation (SCS), pain management, cordotomy, myelotomy, trigeminal nucleotomy, mesencephalotomy, cingulotomy, chronic pain

## Abstract

The aim of this review is to draw attention to neurosurgical approaches for treating chronic and opioid-resistant pain. In a first chapter, an up-to-date overview of the main pathophysiological mechanisms of pain has been carried out, with special emphasis on the details in which the surgical treatment is based. In a second part, the principal indications and results of different surgical approaches are reviewed. Cordotomy, Myelotomy, DREZ lesions, Trigeminal Nucleotomy, Mesencephalotomy, and Cingulotomy are revisited. Ablative procedures have a limited role in the management of chronic non-cancer pain, but they continues to help patients with refractory cancer-related pain. Another ablation lesion has been named and excluded, due to lack of current relevance. Peripheral Nerve, Spine Cord, and the principal possibilities of Deep Brain and Motor Cortex Stimulation are also revisited. Regarding electrical neuromodulation, patient selection remains a challenge.

## 1. Introduction

It is not easy to define pain. Bonica [1] defines it as “an unpleasant sensory and emotional experience associated with actual or potential tissue damage, or described in terms of such damage”. It is an experience imposed on us. It is upsetting. It has to be endured. As Leriche [2] says in his book “Pain Surgery”, only one pain is easy to bear, and that is the pain of others.

Psychiatry, neurology, and other branches of medicine (physiology, anatomy…) have theorized about pain. It has a protective function and favors the survival and evolution of species [3]. It is one of the central themes in the field of philosophy or even theology. It is treated in all religions and widely debated, because it is something inherent and differential in human nature in aspects such as the pain–suffering association or the pain–pleasure dualism [4]. In our efforts to ameliorate or, much better, to control pain, we have to consider all these factors.

From the clinical perspective, three types of pain can be differentiated: 1—symptom; 2—disease [5], showing the characteristics that any disease entails, i.e., chronicity, stability, and alteration of behavior; and 3—complex system. Can it be assimilated to the extrapyramidal system? In this case, each disease is a precise entity, with a different pathogenesis, clinical symptoms, diagnosis, and treatment. Are there, within pain, specific alterations within the structure that integrates pain sensation in the central nervous system (CNS)? Parkinson’s disease, essential tremor, dystonia, and choreoathetosis represent different diseases of the extrapyramidal system; can trigeminal neuralgia, phantom limb pain, or pain caused by a Pancoast tumor or peritonitis be considered to be specific alterations of the pain regulation “system” in the CNS?

## 2. Mechanisms of Pain Modulation

### 2.1. Nociceptors and Peripheral Nerves

A noxious stimulus might cause tissue damage. Physiologically, it results in pain perception by the conscious subject. The condition causing pain often relates to tissue injury (wounds, fractures…), inflammation, traction, etc. At the very onset of perception, the complexity of the pain phenomenon begins.

In the 1930s, Adrian [6] was among the first to analyze the activity of sensory receptors. Impulses through Aδ and unmyelinated C fibers related to painful stimuli. The primary receptor involved in pain perception is a free nerve ending, which transmits information more slowly. However, there are also specific mechanical-, thermal-, or polymodal-sensory nociceptors (which respond to mechanical, thermal, or chemical stimuli) in the skin, viscera, and muscle [7].

The activation of Aδ fibers is associated with rapid, well-defined pain, and C fibers are responsible for secondary pain sensations, which are slow, diffuse, and of longer duration. In general, large fibers have a lower stimulation threshold, both natural and electrical. A stimulus intense enough to activate thin unmyelinated fibers surpasses that needed to stimulate the large fibers, conveying a different pain sensation. Therefore, an elective response to a noxious stimulus in the skin may not appear, transmitting uncertain information about painful stimulation. Several theories have been put forward about this problem of sensory transmission [8]: (1) Theory of the specificity of sensation, by Mac von Frey [9], who stated that each type of receptor transmits information about its specific stimulus; (2) Goldscheider’s stimulus summation theory (1894), in which he proposed that pain is produced by the summation of impulses within the CNS, after applying mechanical or thermal stimuli to the skin; and (3) Nafe’s theory of impulse behavior (1934), i.e., sensation is not based on specific receptors, but on the way in which a given number of impulses are transferred in space (number and type of fibers) and time (frequency). Melzack and Wall [10] incorporated and modified this theory, maintaining an eclectic position, adding the possibility of the existence of stimulus specificity and the absence of stimuli.

The majority opinion was reviewed by Kerr and Wilson [11]. Faced with the alternative of the specificity of pain information or dependence on stimulus behavior, the majority acknowledge a specific system in which nociceptors, with their corresponding afferent fibers, activate different types of neurons in the CNS. Some nociceptors are high-threshold units (responding only to noxious stimuli). But for others, noxious and non-noxious stimuli converge, veiling nociceptor specificity. In this sense, C fibers associated with pain and pleasure have been differentiated: low threshold, tactile, pleasure [12] and high threshold, tactile, pain [13].

### 2.2. Spinal Ganglion-Dorsal Root

At the level of the spinal ganglion, there are two types of cells. The larger ones seem to give rise to myelinated axons, and the small ones give rise to unmyelinated axons. The latter are nociceptive and can be subdivided into two types [14], depending on whether they contain substance P or somatostatin as a neurotransmitter. Randić and Miletić [15] suggest that substance P may play a role as an excitatory transmitter at the level of the first synapse, and that somatostatin may have an inhibitory modulatory effect.

Large fibers converge in a medial tract and penetrate the posterior spinal cords, sending collaterals in small proportion to the dorsomedial area of the dorsal horn, at the level of the substantia gelatinosa of Rolando. The rest end up at the level of Rexed’s laminae IV and V and in the intermediate gray matter (Figure 1).

Aδ fibers and C fibers converge into two bundles: a larger one on the ventrolateral portion of the posterior root and a second, smaller bundle on the dorsomedial surface. Both types of fibers reach the Lissauer tract and are distributed mainly at the level of laminae I, II, and V.

### 2.3. Dorsal Horn of the Spinal Cord

The spinal cord gray matter was initially differentiated by Rexed [17], depending on the cytoarchitecture, into ten laminae, of which the first seven are involved in the pain phenomenon (Figure 1 and Figure 2):Lamina I (marginal layer of Waldeyer) consists of large neurons whose afferents are mostly formed by Aδ fibers and, to a lesser extent, unmyelinated C fibers [18]. They project to the ventral posterior (VP) and intralaminar thalamic nuclei as well as to the periaqueductal gray matter (PAG) [19,20]. These are type I neurons, as described by Iggo [21], responding to painful or almost noxious stimuli.Laminae II and III (substantia gelatinosa of Rolando) contain small-sized neurons receiving most afferents through C fibers, and the rest through collaterals of Aβ fibers of the posterior spinal cord [18]. Their extensions form part of Lissauer’s dorsolateral fasciculus along three or more segments. Their function is still obscure, acting as interneurons and maintaining an inhibitory or excitatory influence on the cells of laminae I, IV, and V [22].The cells of laminae IV, V, and VI (nucleus proprius, neck, or reticular substance and base, respectively, of the dorsal horn) are Iggo type II nociceptive neurons, which respond to low-threshold peripheral mechanical stimuli, but increase their firing frequency when the stimulus becomes noxious [23]. For Mayer and Price [24], the activation of these type II neurons can lead to experiencing pain. From these laminae originates the spinothalamic tract, which projects to the sensory ventral posteromedial and posterolateral (VPM and VPL) thalamic nuclei, as well as PAG.Lamina VII (intermediate zone) gives rise to the spinoreticular tract, next to the spinothalamic tract, and its neurons respond to stimuli that trigger the highest-threshold receptors [25].

### 2.4. Pain Information Pathways

At least three types of ascending pathways transmit pain information to the brain (Figure 1 and Figure 2):

#### 2.4.1. Neospinothalamic Tract or Lateral Spinothalamic Tract

Phylogenetic modifications are needed to improve the direct transmission of pain impulses to conscious levels. Such modifications seem to originate at the level of laminae I, IV, VI, and VII [26]; the fibers cross the midline at the level of the anterior white commissure and ascend through the anterolateral quadrant of the spinal cord. They project, after forming part of the lemniscus medialis, to the level of the sensory thalamic nuclei (VPL, the area of the body; VPM, the area of the face) and from here to the postrolandic sensory cortex, transmitting information about the discriminative aspects of the pain experience.

#### 2.4.2. Paleospinothalamic Tract

It originates in the same type I and II neurons as the previous pathway, but the difference is that they give numerous collaterals to brainstem reticular formation. A multisynaptic pathway is formed that ascends to areas related to behavior (hypothalamus, intralaminar, parafascicular, central medial (CM) thalamic nuclei, and the limbic circuit). It projects bilaterally and seems to be involved in the motivational and affective qualities of pain [7].

#### 2.4.3. Archispinothalamic Tract

Much less differentiated, it appears to be an ascending multisynaptic pathway at the level of the medial medullary reticular substance to diencephalic and cortical zones similar to the paleospinothalamic tract, including the limbic system [27].

In addition, we must consider a descending pain system, which seems to be related to pain inhibition. It was discovered after observing that the PAG stimulation in rats produced analgesia [28], a phenomenon called SPA (stimulation-produced analgesia) [29] that is also seen in humans [30]. This descending inhibitory system, acting on the dorsal horns, is conveyed by the posterior spinal cords and reticulospinal and corticospinal pathways. However, as we will see later, there are inhibitory circuits at the level of each station or relay of the pain pathway (dorsal horn, brainstem, thalamus, etc.).

### 2.5. Brainstem Reticular Formation

It is difficult to obtain a strict scientific definition of the criteria necessary to designate a certain area of the CNS as “reticular”. By exclusion, and unlike well-delimited nuclei and fiber tracts, there are areas where gray and white matter intermingle, fibers form bundles in all directions, and neurons are diffusely distributed, forming poorly defined clusters. Current knowledge of neuronal morphology and connections may not allow us to appreciate a certain organization that actually exists. Most authors include areas located deep at the level of the medulla oblongata, pons, and midbrain as the reticular formation (RF). There is less agreement with regard to including central regions of the spinal cord gray matter, nonspecific thalamic nuclei, and certain hypothalamic nuclei (Figure 2).

The RF is considered to be the oldest phylogenetically, representing the “free neural network” on which more circumscribed and highly organized parts of the CNS have subsequently appeared. However, even the most primitive CNSs have diffuse or highly organized zones. Therefore, it is preferable to look at the RF as the evolution of these more diffusely organized zones or elements, with the following general characteristics: (a) they are deeply located groups of neurons and fibers with a diffuse structural organization; (b) connecting pathways are anatomically difficult or impossible to define, or physiologically complex and often polysynaptic; (c) components of both ascending and descending systems can be recognized; (d) both systems contain crossed and uncrossed elements, eliciting ipsi- and/or contralateral responses after stimulation; and (e) they intervene in somatic and visceral functions

In this complex system, there are massive opportunities for the convergence or divergence of information. In addition, neurons are either excitatory or inhibitory, or cholinergic or aminergic, varying their proportion depending on the region. Importantly, one of the many functions of the RF is short- and long-term homeostasis, with its descending pathways to lower autonomic centers and its ascending pathways carrying visceral and somatic information to the hypothalamus and limbic system. In this sense, we can find centers related to cardiovascular, respiratory, and gastrointestinal control mechanisms as well as to the ascending, bilateral, nociceptive pathways of “slow pain”. Concerning this type of pain, certain regions have been identified (Figure 2):Gigantocellular reticular nucleus and lateral reticular region, at the level of the medulla oblongata and pons—they receive projections from the spinothalamic tract as well as from the deeper areas of spinal cord gray matter [31].Nuclei raphe magnus and PAG, at the level of the midbrain [19,20]—stimulation of the latter in humans causes somatic and visceral sensations accompanied by intense emotional reactions.Ventrolateral PAG and periventricular gray matter (PVG)—lateral to the third ventricle, in the region of the posterior commissure [28], are the areas whose stimulation produces the most effective analgesia with minimal side effects. This SPA [26] may last for several hours after stimulation, suggesting the activation of a multisynaptic system with a mechanism similar to opioid analgesia [32]. Both areas are, indeed, rich in opioid receptors, and their stimulation causes the release of endorphins in CSF. There is, on the other hand, cross-tolerance between SPA and opioid analgesia, just as SPA can be reversed by naloxone [33].

### 2.6. Thalamus

The thalamic structures that directly receive projections of the spinothalamic pain pathway correspond to three groups of nuclei [34] (Figure 2 and Figure 3):

#### 2.6.1. Ventrobasal Complex

The ventrobasal complex includes the VPL nucleus (which receives somatosensory projections representative of the body) and the VPM nucleus (which receives the projections of the trigeminal area). The afferents come mostly from the nuclei of the posterior spinal cord and trigeminal complex, which run along the lemniscus medialis. The other areas projecting into these nuclei are: a) the marginal layer and nucleus proprius of the dorsal horns (and the corresponding area of the spinal nucleus of the trigeminal nerve), whose afferents reach the most caudal part of these nuclei, maintaining a somatotopic representation [19,20]; and b) the mesencephalic reticular substance [35].

Stimulation of the dorsal portion of this complex, in awake humans, elicits non-painful tactile sensations [36]. However, at the level of the caudal and ventral areas, localized pain sensations occur [37]. On the contrary, lesions in the ventrocaudal region produce changes in pain sensitivity that can subsequently lead to hyperpathy [30].

In summary, these VPM and VPL nuclei seem to be involved in the localization and identification of innocuous and noxious stimuli which are preferably conveyed through the lemniscal pathway.

#### 2.6.2. Posterior Complex

The posterior complex comprises an ill-defined area between the medial geniculate body and the pulvinar nucleus. It receives afferents from the nuclei of the posterior spinal cord and the spinothalamic tract, as well as descending cortico-thalamic pathways of the somatosensory cortex. It projects to the retroinsular cortex.

Its stimulation produces unpleasant sensations. The function of this posterior complex in pain is still misunderstood and multiform, and some cells seem to be inhibited by the systemic administration of morphine [38].

#### 2.6.3. Intralaminar Nuclei

Intralaminar nuclei receive afferents from the RF as well as from the marginal layer of the spinal dorsal horn and the spinal nucleus of the trigeminal nerve. They receive virtually no projections from the lemniscal pathway. Like the latter complex, they receive ipsi- and contralateral information [39]. Their stimulation causes diffuse unpleasant sensations and, sometimes, poorly localized pain. A lesion produced at this level can relieve incoercible pain [30], and cells at this level can also be inhibited by the administration of morphine.

### 2.7. Cerebral Cortex

The cerebral cortex constitutes a higher level of the pain pathway, and three different somatosensory areas can be differentiated and highlighted (Figure 3).

#### 2.7.1. Primary

The primary area corresponds to the postrolandic gyrus, including its medial extension into the paracentral lobe. It receives projections from the VPL and VPM nuclei. Stimulation produces well-localized sensations of touch and temperature, but rarely pain [40]. Its removal may result in a loss of sensory-discriminative capacity, and sometimes improves pain, although hyperpathy may appear later [41].

#### 2.7.2. Secondary

Secondary areas are located at the level of the superior operculum of the Sylvian fissure, behind the fissure of Rolando and the retroinsular region. Many of the cells in these areas respond to noxious stimuli, and their stimulation causes localized pain sensations. There are reciprocal projections to the thalamus, VPM, VPL, and medial nuclei. Lesions at this level alter pain sensibility, not affecting the somatosensory-discriminative capacity [42].

#### 2.7.3. Tertiary

Tertiary areas correspond to the areas of the anterior cingulate and insula. They are related to the limbic system and visceral sensations. They connect with the medial thalamic nuclei and are related to the affective-emotional component of pain [43].

Cortical resections have been abandoned because of their failure to relieve pain and subsequent complications. Only cingulotomy has remained as an analgesic intervention, although its effects on pain are due more to its action on personality and the affective component of suffering rather than on pain as a sensory activity [41,44]. The function of the cerebral cortex in the pain phenomenon remains elusive.

### 2.8. Pain Modulation Pathways

#### 2.8.1. Neurophysiological

Experience indicates that chronic pain is rarely fully abated, rapidly or lastingly, after some type of surgical, physical, or behavioral intervention. Usually, after an appropriate regimen, patients might feel less pain and accordingly change their behavior in the long term, enabling them to lead a more productive life.

Numerous efforts have been made to explain this ability to modulate and modify pain. Melzack and Wall [10] put forward a theory that persists today, although adapted, to explain these phenomena at the level of the first relay of the pain sensory system. The following provides a brief explanation of the gate control theory (Figure 1 and Figure 2):

At the spinal cord segmental level, there is a dynamic interaction between the input of information through the large myelin fibers (A) and that of thin (Aδ) and unmyelinated (C) fibers. This interaction takes place at the level of laminae II and III (substantia gelatinosa of Rolando [SGR]), where there are neurons that exert a presynaptic inhibition of both types of fibers at their synapses with T (transmitter) cells, presumably located in lamina V, from which the spinothalamic fibers arise directly or indirectly.

On the one hand, the large fibers send collaterals that excite the interneurons of the SGR, enhancing the inhibition of the T cells (the gate is closed, preventing the passage of pain information). On the other hand, the unmyelinated fibers send inhibitory information to SGR cells, reducing T cell inhibition (thus opening the gate, allowing the passage of pain information).

Melzack and Casey [45] modified this behavioral approach with another theory, i.e., “central control trigger”, which implied that the information transmitted by the large fibers through the posterior spinal cord can activate central inhibitory mechanisms, whose downstream action modulates pain transmission by dorsal horn neurons. T cells can be inhibited or excited by information from the brainstem, which uses the posterior spinal cords as a transmission pathway [21]. Melzack [46], in 1975, further suggested that certain forms of neurostimulation activate a central biasing mechanism that inhibits chronic and pathological pain signals. This mechanism, with a neurophysiological or neurohumoral component, delays the pathological processes of fixation and correlation with old experiences, which occur simultaneously with the reception of the pain signal (in which the sensory inputs, inhibitory pathways of the brainstem, activity of the autonomic nervous system, individual’s expectations and anxiety, personality structure, etc., are combined).

However, just as the existence of a modulatory system at the level of the first relay in the spinal dorsal horn is becoming better known, there are, nevertheless, more complex and less understood modulatory systems located at higher levels of the CNS. As an example, we will only cite the fact that stimulation at the level of the VPL nucleus [47] or the posterior limb of the internal capsule (thalamocortical pathway, of specific afferents) [48] can produce analgesia. This phenomenon is explained by an inhibitory effect of these structures on other thalamic nuclei or the cortex [35,49]. This descending inhibitory system involves structures such as the ventrolateral prefrontal cortex [50], the parahippocampal region [51], or the amygdala [52]. Therefore, chronic pain could be the result of a failure of inhibitory mechanisms rather than an excess of inputs [53].

#### 2.8.2. Neurochemistry

In 1975, Hughes et al. [54] isolated two endogenous pentapeptides of morphinomimetic action: methionine and leucine–enkephalin. Subsequently, larger peptides, endorphins, with the same action and at many levels of the CNS were identified [55,56]. PAG and PVG are two areas with the highest abundance of endogenous opioid receptors, and in which the analgesia produced by their stimulation (SPA) has behavioral consequences similar to those obtained after the administration of morphine.

Enkephalins and endorphins have many potential places where they either modulate or serve as intermediaries by activating other systems (Figure 4). As an example, there are enkephalinergic neurons at the level of the marginal zone and SGR; these same neurons also apparently receive many projections of primary nociceptive afferent terminals, whose neurotransmitter is substance P. Both morphine and enkephalins inhibit the release of substance P and reduce the response of neurons to the pain stimulation of their respective peripheral organs. Both morphine and enkephalins can act at this spinal cord level, activating a descending bulbospinal serotonergic system starting at the level of the raphe nuclei, which receive their information from the PAG. Both electrical stimulation of the PAG and administration of opioids at this level will produce the same analgesic effect, which can be blocked by the systemic administration of naloxone or by lesions, either at the level of the raphe nuclei or of the descending pathway in the dorsolateral fasciculus of Lissauer [57].

However, the mechanisms are even more complex because, for example, the same nociceptive impulses can (through collaterals to the gigantocellular reticular nucleus and from here to the PAG) inhibit (via the lateral dorsal funiculus) the peripheral impulses reaching laminae II and V cells [58]. Therefore, there are negative feedback mechanisms, which are mediated by endorphins and use serotonergic pathways.

### 2.9. Gate Control Theory

Perhaps it would be better to have a simpler idea of what can occur at each relay of the pain pathway (dorsal horn, brainstem, thalamus…), helping to clearly explain the positive effects of surgical treatment.

In this sense, we can imagine that both types of sensory (S) and pain (P) information arrive at higher brain levels (Figure 5). If, for any reason, the latter increases, the gate will turn to the left, preventing the passage of sensory information (we pay more attention to pain, i.e., to the nociceptive stimulus that has occurred).

If this information were to remain constant and become chronic, we could reduce it in two ways (Figure 6), either by inhibiting the afferents of the same pain pathway (by pharmacological, instrumental, or even surgical means) or acting on the sensory pathway by increasing its afferents. The latter can be done in a natural way (for example, rubbing the skin after receiving a blow dulls the pain) or by means of neurostimulation.

Another way of understanding the problem is the model proposed by De Ridder, et al. [4] “in which pain (and suffering) is the consequence of an imbalance between the ascending and descending pain inhibitory pathways. This balance is theorized to be under control of the reward system”.

## 3. Pain, from the Clinical Point of View

### 3.1. Types of Pain

#### 3.1.1. According to Duration: ACUTE or CHRONIC

Acute pain is physiological. It warns of tissue damage and is transient and reversible. It produces coordinated defense responses, such as the withdrawal reflex, as well as autonomic responses such as increased heart rate and blood pressure, with activation of the hypothalamic-pituitary axis, while limiting mobility in order to promote tissue repair.

Pain can become chronic if the noxious stimulus is maintained over time or by alteration of the mechanisms involved in the onset and resolution of pain. Therefore, it perpetuates itself, not as a symptom of the cause or disease that produced it but as a disease in itself, even if the problem that generated it has been solved. Bonica [8] establishes a period of 3 months to differentiate between acute and chronic pain. Pain becomes an entity of its own and not a mere symptom of a disease [59].

There are genetic, psychological, and even environmental factors that can favor the transition from acute to chronic pain [4,60].

One of the best clinical descriptions of acute and chronic pain is that of Hendler [61], distinguishing four stages:Acute—Up to 2 months from onset

From the clinical point of view, this is “normal” pain. It does not involve any type of psychological disorder. It responds to analgesics. Individuals who experience this pain hope to be cured.

b.Subacute—From 2 to 6 months from onset

Individuals who experience this pain begin to exhibit distress, somatizing pain. Anxiety and irritability appear, along with insomnia and social isolation. Individuals become increasingly concerned about pain, taking on an important role in daily life. Sexual activity decreases. These individuals take sedatives and analgesics and wait and trust that the pain will subside.

The pathways in both stages are those of physiological pain, from ongoing nociception (Figure 1 and Figure 2).

c.Chronic—From 6 months to 8 years

Individuals present with depression and anguish and slowly become hostile toward doctors and caregivers, but remain dependent on their treatments, although having experienced therapeutic failures. These individuals progressively abandon normal family and social life and abandon employment.

This distress may be related to the activity of the rostral part and anterior dorsal surface of the corpus callosum [62], insula and nucleus accumbens [63] as well as the dorsolateral prefrontal cortex [64].

d.Hyperchronic—3 to 12 years

Individuals who experience this pain learn to live with it, but do not accept it and cease treatment, because doing so confirms lack of efficacy and irrelevance, allowing a decrease in possible side effects. Individuals recover their family and social life and return to work. Depression and insomnia improve.

The latter two stages involve the paleospinothalamic and reticular substance pathways, with involvement of the hypothalamus and the limbic system [65] (Figure 1, Figure 2 and Figure 4).

#### 3.1.2. According to the Mechanisms Involved: NOCICEPTIVE and NEUROPATHIC

The first occurs as an effect of a nociceptive stimulus (mechanical, chemical or thermal) on the receptors. Acute pain is always nociceptive. It serves to protect the injured region, until it heals [66]. In theory, there is no injury or malfunction of the complex system supporting pain as a physiological function.

The second is due to a lesion in the peripheral nervous system or CNS that damages the dynamic structure and, consequently, the balance, sustaining pain as a “system” similar to the extrapyramidal system, as mentioned earlier. One of the main features is that normal sensory afferents are diminished or abolished. Other names are used to define the same type of pain, causing confusion. For example, deafferentation pain defines the main characteristic of the pain; a paradigmatic example is phantom limb pain. Another term is central pain, when the injury causing the pain affects CNS structures; paradigmatic examples of this condition are Dejerine Syndrome and thalamic pain [67].

#### 3.1.3. According to Life Expectancy: BENIGN and MALIGNANT PAIN

Pain can be classified on the basis of the characteristics of the disease that produces it, depending on whether the underlying condition is benign or malignant (cancer). This differentiation is very important in planning and providing the best treatment, having achieved long life expectancies in recent times.

### 3.2. Pain from Ongoing Nociception

All tissues (except the CNS) are innervated by nociceptors, responsible for locating and quantifying the intensity of stimuli that can cause tissue injury (trauma, infection-inflammation, visceral strains, neoplastic infiltration, etc.). In addition, there are thermoreceptors and mechanoreceptors that also give warning signals after reaching a certain threshold.

In addition to physical stimuli, nociceptors also respond to the extracellular chemical environment altered by the cellular lesions caused by these stimuli, especially to inflammatory mediators (bradykinin, histamine, serotonin, etc.). This can lead to an abnormal response by nociceptive neurons: (a) allodynia, when reacting to stimuli below the pain threshold; and (b) primary hyperalgesia (increased pain sensation above what corresponds to the painful stimulus received) [7].

The main nociceptive pathways (Figure 2) can be summarized as follows:

#### 3.2.1. Ascending Pathways

##### Sensory-Discriminative Component of Pain

Ia.Spinothalamic
Pain-heatSomatotopic distributionIb.Posterior spinal cords
Touch, vibration, discriminationProprioceptionSomatotopic distributionII.Thalamus
VPM, VPL, and Po (Posterior Complex)III.Primary and secondary sensory cortex

##### Affective-Emotional Component (Unpleasant)

Ia.Spinoreticular tract
Medullary reticular substanceLaminae V-VIIIDoes not localize painActs on autonomic mechanismsIb.Spinomesencephalic tract
PAGSomatotopic distributionIc.Ventral spinothalamic tract
VPL and Po (Posterior Complex)Id.Multisynaptic ascending systemII.HypothalamusIII.Medial thalamusIV.Limbic system

#### 3.2.2. Descending Pathways

With modulatory, inhibitory action:Ia.Posterior spinal cords
Cerebral cortexDiencephalonIb.Spinoreticular tract
Originates from the PAG

#### 3.2.3. Examples of Chronic Pain Syndromes from Ongoing Nociception

Low back pain (failed back syndrome)Ischemic pain (angina pectoris)CoccygodyniaMyofascial syndromes (fibromyalgia)Postoperative (thoracic surgery)Cancer pain…

### 3.3. Neuropathic Pain

From a clinical perspective, there is a type of chronic pain that does not meet the canons of pain from ongoing nociception. This type of pain is differentiated by pain characteristics that have a previous antecedent clearly related to a previous disease that has directly affected some nervous system structures.

However, there is no consensus regarding the terminology for this type of pain, with three preferred terms: neuropathic pain, deafferentation pain, and central pain.

Yezierski [68] describes the historical sequence of central pain very well. The first allusions to this type of pain appeared about 200 years ago, in relation to Wallenberg syndrome [69] and other vascular affectations at the level of the brainstem and thalamus. Dejerine’s work in 1906, which defines thalamic syndrome, is a classic reference. The term central pain appears for the first time in 1914 [70,71,72].

The characteristics of neuropathic pain are persistence, spontaneous onset, intermittent, lancinating, and burning, accompanied by hyperesthesia, hyperalgesia, allodynia, and hyperpathy, with exalted motor and autonomic reactions. Two different targets or injuries within the nervous system were differentiated: peripheral nerves and CNS structures (spinal cord–encephalon). The first was called neuropathic pain and the second central pain [73]. Thus, in 1994, the International Association for the Study of Pain (IASP) defined central pain as that associated with a primary injury or dysfunction of the CNS [74,75].

Tasker et al. [76], in 1980, proposed the term deafferentation pain for pain associated with an injury to nervous system structures, ranging from the spinal roots and plexuses (anesthesia dolorosa, phantom limb pain, postherpetic neuralgia, brachial plexus avulsion, etc.) to spinal cord and brain injuries [77,78].

Some authors prefer to separate pain caused by peripheral nervous system injuries from that caused by CNS injuries [79].

Although, today, the concept of neuropathic pain predominates, confusion or simultaneity with the concept of deafferentation pain continues [80].

Notably, some deafferentation pain syndromes are caused by surgical techniques for pain relief themselves (cordotomy, mesencephalic tractotomy, thalamic lesions, etc.).

It is impossible to achieve unanimity in classifying and naming the numerous syndromes included in the non-nociceptive pain spectrum, although three names remain useful: central, neuropathic, and deafferentation pain.

Perhaps the term neuropathic pain should be used more for syndromes with involvement of the peripheral nervous system, and the term central pain should be used for those with CNS involvement. The more complex term of deafferentation should be left as a detrimental adjective for both, when there is a clear lack of sensory afferents.

Unlike pain from ongoing nociception, neuropathic pain and central pain do not respond to opioids [81]. Drugs that increase dopaminergic transmission have some effect, as do different antiepileptic drugs because of their GABAergic action (clonazepam, valproic acid, etc.) or antiglutaminergic activity (lamotrigine)) and antidepressant drugs [68].

Examples of neuropathic and central pain syndromes:Postherpetic neuralgiaBrachial plexus avulsionPhantom limb painReflex sympathetic dystrophyCausalgiaSpinal cord injuryThalamic syndromeCancer pain…

### 3.4. Other Pain Syndromes

Craniofacial pain encompasses a large number of entities that may not fall into the above classification. These include the following:Headaches and migrainesEssential trigeminal neuralgiaSecondary trigeminal neuralgiaAtypical facial neuralgiaGlossopharyngeal neuralgiaArnold’s suboccipital neuralgiaOther neuralgiasOrofacial pain…

The aim of this review excludes the analysis of this type of pain.

## 4. Neurosurgical Procedures

### 4.1. Introduction

Having reviewed the main concepts of the pathophysiology and clinical presentation of intractable or uncontrolled pain even with adequate medical treatment, it is now important to consider the main surgical procedures and their indications, results, and adverse effects in order to provide a clear understanding of the surgical possibilities.

Such knowledge allows a more comprehensive view of the treatment options offered to a given patient, from the most appropriate medical treatment to anesthetic techniques and to surgical approaches.

If this global vision is maintained, patients will have access to a stepwise and multidisciplinary treatment plan from the very beginning.

### 4.2. History of Pain Neurosurgery

Because of its relevance and frequency, trigeminal neuralgia stands out in the history of surgical pain treatment.

In 1891, Horsley [82] proposed avulsion of the Gasserian ganglion for the treatment of trigeminal neuralgia. The techniques for this intervention were improved by Cushing and later by Frazier. This type of intervention was performed until the 1970s, with selective sections of one or more trigeminal branches at the level of the Gasserian ganglion.

In 1925, Dandy [83] proposed the complete or partial section of the Trigeminal root using a suboccipital approach. This intervention continued until the 1990s [84]. Even today, some neurosurgeons propose selective sections for cases in which there is no clear vascular compression.

In the mid-1940s, Spiegel et al. [85] built and applied the first stereotactic guide for the treatment of various CNS conditions, such as epilepsy, psychotic disorders, and pain, by lesioning different targets in the thalamus.

Edgar Moniz, in 1949, received the Nobel Prize for his proposal for the surgical treatment of serious psychotic disorders, coining the term psychosurgery.

Closely related to this field are the proposals for the treatment of intractable pain by means of lesions in different cortical and subcortical targets [86,87]:Spiegel and Wycis [85]—dorsomedial thalamotomyTalairach and Leksel [88]—anterior limb of the internal capsuleScoville [89]; Whitty [90]—anterior cingulotomyKnight [91]—subcaudal tractotomyKelly [92]—limbic leukotomy

At a lower level, the following proposals for ablative procedures stand out:Abbe [93]—spinal rhizotomyLeriche [94]—sympathectomySpiller and Martin [95]—cordotomyArmour [96]—myelotomySpiegel and Wycis [97]—mesencephalotomy

Regarding neuromodulation procedures, the following milestones should be highlighted:Richardson and Akil [30]—stimulation of PAG and PVGHosobuchi et al. [32]—stimulation of sensory thalamic nucleiPenn and Paice [98]—intrathecal morphine

Next, we will review the most important neurosurgical techniques, both ablative and neuromodulatory, with their indications, results, and adverse effects.

### 4.3. Ablative Techniques

These approaches are not in vogue because they have several disadvantages: (A) morbidity and complications inherent to the surgical procedure; (B) onset of new deficits, which can lead to the appearance of new deafferentation pain; and (C) long-term pain recurrence in benign painful conditions [80].

In malignant pain, which does not respond to treatment with opioids, even by intrathecal administration, an ablative approach can be proposed, always considering life expectancy. Therefore, patients with a life expectancy of less than 3 months should not be considered candidates because of the high risk of morbidity/mortality [99]. At the other end of the spectrum are patients with a long life expectancy; they can be assimilated to benign pain, for which an ablative procedure can cause deafferentation pain that arises approximately two to three years post-treatment.

#### 4.3.1. Peripheral Neurectomy

There is currently no indication for this procedure. It causes sensory and motor deficits, can lead to greater deafferentation pain, and can even cause a neuroma.

Another situation is pain caused by a direct injury to a peripheral nerve. Treatment of a painful neuroma or nerve release may be indicated.

In Arnold’s suboccipital neuralgia, this alternative could be considered in very select cases, in which entrapment is suspected and only a nerve release is intended.

#### 4.3.2. Rhizotomy and Ganglionectomy

Abbe [93], in 1896, proposed the intradural section of the dorsal root. Scoville [100], in 1966, proposed an extradural section. Later, Uematsu [101], in 1974, used a percutaneous technique to perform radiofrequency rhizotomy.

Open surgical techniques have been abandoned. They have been replaced by percutaneous selective radiofrequency ablation techniques.

The best results are achieved for malignant pain involving the brachial plexus or invading the chest wall, as well as pain in the pelvic region. For chronic benign pain, select cases of suboccipital neuralgia, thoracic fractures, and failed back syndrome may be relieved [102].

These procedures are disadvantageous because of the likelihood of recurrence and the risk of causing long-term deafferentation pain.

#### 4.3.3. Sympathectomy

Leriche [2], in his book on the treatment of pain, proposes a whole body of doctrine on this type of surgical approach. Subsequently, the different sympathectomy techniques have come to be performed in a less invasive way, falling within the field of percutaneous techniques, with indications for conditions such as causalgia, reflex sympathetic dystrophy, and vascular-ischemic disorders, although without clear scientific evidence of their results [103].

#### 4.3.4. Dorsal Root Entry Zone (DREZ) Lesion

This area was first described by Sindou [104,105,106]. This region includes the medial portion of the dorsal root, Lissauer’s tract, and laminae I to V of the dorsal horn (Figure 7). This zone is the first pain relay, where excitatory and inhibitory actions converge in the transmission of the pain signal.

The lesion of this zone (with bipolar microcoagulation), in conditions in which there is a clear injury (brachial plexus avulsion, for example), can lead to restabilization in this area.

Nashold et al. [107,108] modified the technique using radiofrequency. Sindou et al. [109] improved the surgical approach, adding intraoperative neurophysiological monitoring.

This is the only ablative procedure that can be considered to treat deafferentation pain—in particular, brachial plexus avulsions, other plexus disorders (Pancoast syndrome), spinal cord injuries (with clearly metameric pain), causalgia due to peripheral nerve injury, and hyperspasticity with pain. In postherpetic neuralgia, allodynia is controlled, but burning and prickling sensations remain.

#### 4.3.5. Cordotomy

The spinothalamic tract was first described by Spiller and Martin [95] in 1912. Subsequently, percutaneous techniques were developed, decreasing morbidity [110].

The main indication for cordotomy is malignant pain, at the level of the lower body and preferably unilateral. The main disadvantage is the occurrence of deafferentation pain approximately two years after treatment in up to 20% of patients (Figure 8).

The administration of opioids has diminished or almost eliminated this technique, although recent reviews of published series demonstrate its efficacy [111], all showing good results. There is even mention of increased efficacy of intrathecal morphine [112] as a result.

#### 4.3.6. Myelotomy

##### Commissural Myelotomy

In more central pain, bilateral anterolateral cordotomy poses many adverse effects, which increase if performed at the cervical level. Therefore, one possibility is a medial incision in the spinal cord to sever the crossing of the spinothalamic tract. This was first performed by Armour [96] in 1927, and by Leriche in 1928 [2].

The idea was to sever not only the posterior white commissure, but also the anterior white commissure [113]. Although it has good results for pain, there are numerous side effects, such as gait disorders, deep sensitivity disorders, posterior cord involvement, or paresis; however, as in bilateral cordotomy, no sphincter disorders were reported [114] (Figure 9).

##### Extralemniscal Myelotomy

In order to control pain in the upper body involving the neck and upper extremities, Hitchcock [115,116] proposed percutaneous thermal lesion at the cervical level (C1), at a depth of 5 mm from the posterior spinal cord surface. Given the ability to relieve pain well below the lesion, the author hypothesized that this lesion severed a multisynaptic medial pathway different from the spinothalamic tract. The same positive results were published by Schvarcz [117] in 1984.

These procedures have fallen into disuse due to complications and the effectiveness of opioid treatment. However, this approach should be considered for its possible beneficial effects on pelvic visceral pain that is unresponsive to pain control, because myelotomy at the thoracic level can produce a good result without adverse effects [118] (Figure 9).

#### 4.3.7. Procedures at the Level of the Brainstem

##### At the Level of the Trigeminal Complex

There are three types of lesions, very similar or in continuity:Trigeminal tractotomy—The trigeminal tract also carries nociceptive afferents from cranial nerves VII, IX, and X. Sjöqvist [119] proposed in 1938 that this tract be severed, just before entering the caudate nucleus. The procedure involves a transverse section, about 8 mm below the obex (Figure 10). It generates ipsilateral hypoalgesia in the face, mouth, pinna, back of the tongue, and pharynx. Hitchcock [115,116], in 1970, described a percutaneous technique.Trigeminal nucleotomy—Schvarcz, since 1970, has sectioned the superior part of the caudate nucleus. The inferior caudate nucleus continued with the spinal dorsal horn at the cervical level [120]. (Figure 11).Caudate nucleus-DREZ lesions—This approach involves a lesion in the gelatinous substance of the caudate nucleus. It was proposed as an open surgery by Nashold et al. [121] in 1992. It involves the destruction of the nucleus from C2 to 5 mm above the obex (Figure 12).

Techniques have been described by El-Naggar and Nashold [123] in 1995 and Gorecki and Rubin [124] in 2002.

These interventions are very similar to the original DREZ lesion proposal. They are indicated in the treatment of severe deafferentation conditions, such as facial anesthesia dolorosa, postherpetic facial neuralgia, and orofacial cancer pain [125].

Young [126] noted that in his experience, this technique should be reserved only as a last resort, and should be performed only by experienced hands, given the possibilities of complications and the low rate of excellent pain outcomes.

##### Stereotactic Mesencephalotomy

The aim of this approach is to sever the afferents in the spinothalamic tract above the upper limb and face, as if it were a superior cordotomy [127].

At first, the approach had unacceptable morbidity and mortality rates. The technique was later refined by a stereotactic approach [97].

There was a transition to a lesion of the paleospinoreticular pathway rather than the spinothalamic tract (Figure 13 and Figure 14), because the most common and disabling complication of significant dysesthesias resulted from the lesion of the nearby lemniscus medialis.

Spiegel et al. [130], in 1954, showed in cats that the stimulation of the periaqueductal reticular substance produced emotional responses to pain, and suggested interrupting the extralemniscal reticular substance could be beneficial. In 1964, this author proposed the lesion of the medial thalamic nuclei, where the extralemniscal reticular polysynaptic pathway is projected.

Nashold [128] proposed a lesion in the lateral zone of the PAG and the medial RF with respect to the spinothalamic tract, i.e., extralemniscal mesencephalotomy, with the aim of also treating the emotional component of pain. Amano [131], in 1998, proposed that the lesion be produced at a somewhat lower level, in order to avoid injuring the oculomotor nuclei.

The main indication is uncontrollable oncological pain at the orofacial level [132].

#### 4.3.8. Cortical-Subcortical Lesions

##### Medial Thalamotomy

This approach was proposed by Spiegel et al. [85] in 1947. Some authors, such as Gildenberg [129,133], prefer it to mesencephalotomy. One of the proposed targets is the central-lateral intralaminar nucleus [134]. Young et al. [135] proposed, in 1995, the use of a gamma knife to produce the lesion (Figure 5).

Tasker [136] believes that this technique, although having less morbidity than mesencephalotomy, is less effective for nociceptive pain.

##### Cingulotomy

Cingulectomy refers to the resection, in open surgery, of the anterior 4 cm of the cingulate gyrus (Brodmann area 24). It disrupts connections, including the cingulate fasciculus. Le Beau [137] was the first to perform this technique.

Cingulotomy involves sectioning the cingulate fasciculus, located below the cingular cortex [138]. It has connections with the frontal lobe, temporal lobe (through the uncinate fasciculus), and the limbic system. The first intervention under stereotaxic control was performed by Foltz and White [139].

The main objective is to correct the affective disorder associated with chronic pain (anxiety, depression, suffering, and emotional lability).

Cingulotomy must be bilateral to achieve the best results. However, it has no effect on nociceptive somatic perception [140]. If the result is good, even if transient, the intervention may be repeated to enlarge the lesion [141].

Cingulotomy may be an option to consider for patients with malignant pain, with orofacial involvement, and with affective component [142]. Patel et al. [143] proposed cingulotomy using new technologies such as lasers and frameless stereotaxis.

A more recent proposal involved the bilateral stimulation of the ventral and dorsal areas of the anterior cingulate cortex [144].

##### Hypophysectomy

This approach was proposed to treat pain in patients with multiple metastases [145]. The aim was to slow the progression of metastases by modifying hormone levels [146]. In 1995, Talairach and Tournoux [147] proposed hypophysectomy after the implantation of radioactive seeds.

In the 1970s, hypophysectomy as a treatment for pain in patients with breast or prostate cancer with disseminated metastases became more common [148].

Advances in the medical treatment of pain have rendered this technique obsolete since the 1980s [149]. Even so, the possibility of performing this technique successfully via radiosurgery should be considered [150,151].

### 4.4. Analgesic Neurostimulation Techniques

#### 4.4.1. Historical Introduction

This therapeutic approach has become attractive because there is no definitive injury, and therefore, the new situation created is reversible at any time. Although this approach has not yet achieved, in general, the same effectiveness as lesioning techniques, the efforts to improve the knowledge and pathophysiological foundations of analgesic neurostimulation, along with technological advances and improvements in the form of application, have greatly boosted a therapy that was probably one of the first to be applied in the origins of medicine.

The ancient Egyptians knew that some fish produced electric shocks (electric rays, eels, etc.), and they used the numbing effects therapeutically [152]. In Greek, the name for the electric eel, *νάρκη*, is derived from the verb *ναρκάω* (to numb, to paralyze) and the term *ναρκωσις* (narcosis). These were mentioned in the writings of Hippocrates and Galen.

However, there is no written record of the use of electric shocks from these fish until 46 AD, in which Scribonius Largus recommends them for the treatment of headaches and gout. This was the first written reference to the use of transcutaneous neurostimulation for the treatment of pain. Scribonius Largus reported that the effect of shocks was slowly progressive and that numbness could persist even after contact with the fish was interrupted. This numbing effect has been described throughout the centuries in different medical treatises and has even been applied by certain primitive tribes [152].

From the time William Gilger (1544–1603), an English court physician, first described magnetic phenomena and coined the term electricity, it took a century to invent the first machine, thanks to Otto de Guericke, capable of producing static electricity by rubbing a sphere of sulfur. It took another 100 years for the appearance of electrostatic machines for medical purposes, constructed by Kratzenstein and Krugger in 1744 and Jakob Hermanna Klyn in 1746. Richard Lovett, in 1756, published the book “Subtil Medium Proved”, which was the first on medical applications of electricity. John Wesley, the founder of Methodism, was at this time the great defender of electrotherapy, describing its indications in the book “The Desideratum”, published in 1769, including headaches, sciatica, pleuritic pain, angina pectoris, and ischemic pain as processes that could be relieved with this therapy. At the end of the eighteenth century, the use of electricity declined because expected results were not achieved, and electrotherapy fell into disregard.

Galvani’s discovery and, above all, Volta’s invention of the electric battery in 1800 caused a revolution in physics and gave new impetus to the development of electromedical devices. In this field, Duchenne De Boulogne [153], who acquired great experience and differentiated the physiological effects and applications of the different types of electric currents, stands out. Later, Hermel, in 1844, cited his experience in electropuncture, and Sarlandiere applied electric current to acupuncture needles. Thus, the indications of electrotherapy were more clearly defined, making it the most important treatment for pain [154].

However, enthusiasm waned, and therefore, at the beginning of the twentieth century, only attempts to pass electricity through the brain, aiming to produce narcotic effects, stood out. However, with the exception of electroconvulsive treatment (electroshock) described by Cerletti and Bini [155] in 1938, electrotherapy did not occupy a prestigious place in medical therapeutics during the first 70 years, although its application was maintained at the basic research level [156].

More recently, neurostimulation has again experienced an unusual boom, most likely due to two fundamental facts [157]. First, Hess [158], in work published in the 1930s, drew attention to the possibility of producing motor and emotional manifestations by means of electrical brain stimulation. Second, in 1965, Melzack and Wall [10] published the gate control theory on pain, which immediately produced therapeutic applications. Wall and Sweet [159], in 1967, reported the positive results of electrical stimulation at the level of the peripheral nerve trunks. This technique is based on the theory of blocking pain afferents by means of electrical or tactile stimulation, which activates Aβ fast-conducting fibers. Almost simultaneously, Shealy reported cases of electrode implantation at the level of the posterior spinal cord, where the highest concentration of fast-conducting myelinated fibers is found.

After observing that stimulation preceding the lesion of the sensory thalamic nuclei (VPL-VPM) produced paresthesias, Mazars et al. [160], in 1960, reported a sustained beneficial effect, or even cure, obtained after stimulating only for a few days. Reynolds [28], in 1969, described the analgesic effect of the stimulation of the central gray matter in rats, and in 1977, Richardson and Akil [30] reported the first electrode implantations in humans, at the level of the PAG and PVG. The discovery of morphine receptors at the brain level [161] and endogenous peptides with morphinomimetic action gave a new dimension to neurostimulation [32,162].

Neurostimulation, also known as neuromodulation and neuroaugmentation, began to be applied, with greater or lesser success, not only as a therapy for certain types of intractable pain but also for other types of problems, such as spasticity [163] and epilepsy [164].

For the treatment of pain, analgesic neurostimulation emerged with the aim of reducing pain by interfering with its two fundamental regulators, tactile and proprioceptive transmission, and higher inhibitory mechanisms [16]. The application of electrical stimulation in certain areas of the nervous system is achieving clear, positive results. Its indications are becoming more clearly identified due to the greater knowledge of neurophysiology, as well as to undoubtable technological advances. Its medium-term future remains promising, in terms of offering a nonaggressive alternative to modify the response of the nervous system itself to pain. However, it is impossible to predict whether other types of therapy (mainly pharmacological), which are also undergoing considerable progress, will fail to overcome or modify this therapeutic alternative.

#### 4.4.2. Transcutaneous Electrical Nerve Stimulation

Transcutaneous electrical nerve stimulation (TENS) [165] involves passing an electrical current through the skin to modify nervous system responses. Because an analgesic effect is intended, the target structures on which it aims to act are large sensory fibers of the peripheral nerves.

There is still some skepticism about the use, safety, and effectiveness of TENS. However, the electric current generated is extremely safe and controllable, having sophisticated systems that have passed tests required by organizations such as the American FDA and the European Community; thus, its application is restricted and controlled by a medical practitioner.

The electrical stimulus is radiated through the skin until it reaches the nerve structure, increasing the depolarization of the membrane at the level of the negative electrode. The first fibers affected are large, fast-conducting myelinated fibers. Increasing the intensity of stimulation may trigger an action potential in the highest-threshold slow-conducting unmyelinated fibers that convey pain sensations. This is counterproductive, and the energy transmitted to the patient must, therefore, be adapted by choosing a suitable stimulus for the fibers intended to recruit, which, according to the gate control theory, are large myelinated afferent fibers.

The territories to be stimulated can be (a) at the level of the painful area, looking for a trigger point for the pain, using the Travell and Rinzler tables cited by Melzack et al. [166] as a reference; (b) a nerve path that collects sensory information from the painful area [165]; and (c) acupuncture points corresponding to different charts [167].

##### Mechanism of action of TENS

There are several theories, which are summarized below:Production of a peripheral blockade of the painful stimulus, using an antidromic mechanism [168];“Gate control theory” by Melzack and Wall [10];Theory of the “central control trigger” by Melzack and Casey, which implies nociceptive information ascending through the posterior spinal cords and triggering a central inhibitory mechanism that, downstream, acts at the level of pain stimulus entry into the dorsal horn;Reducing the sympathetic tone [169]; andAn action mechanism similar to acupuncture, as described by Fox and Melzack [167].

##### Indications and Results

TENS was originally designed as a method to predict the success or failure of implantable stimulators. It was soon accepted as a treatment modality for numerous syndromes and types of pain [170,171].

The applications are manifold and were thoroughly analyzed by Manheimer [172]. (a) For acute pain, TENS can, in many cases, replace medication and assist in reducing the need for drug treatment [173]. (b) For different types of chronic pain, TENS may be the first modality to try, together with appropriate psychological and physical therapy measures [133]. In general, TENS provides a long-term benefit of no greater than a 50% reduction in pain, decreasing to 33% when used as the only therapy [171].

Gibson et al. [174] stated the following: “We were therefore unable to conclude with any confidence that, in people with chronic pain, TENS is harmful, or beneficial for pain control, disability, health-related quality of life, use of pain relieving medicines, or global impression of change”.

#### 4.4.3. Peripheral Nerve Stimulation

This modality was first used by Wall and Sweet [159]. Its principle of action is similar to TENS, but a surgery is required for the implantation of both the electrodes and the subcutaneous stimulator.

Its application spread relatively quickly [175,176]. However, the fibrosis produced causes a progressive loss of the analgesic effect, requiring an increase in the intensity of the stimulus, with risks of nerve damage [177].

Its indication is restricted to neuropathic pain due to localized and incomplete traumatic lesions of peripheral nerves which respond well to percutaneous tests, with approximately 50% of patients reporting good results [178,179]; however, no randomized studies have been conducted.

Possible complications include infection, greater surgical difficulty than other noninvasive techniques, and the possibility of perioperative nerve damage or subsequent fibrosis.

Within this type of treatment, two modalities with acceptable results should be considered. The first is chronic Gasserian Ganglion stimulation for atypical facial pain [180,181]. No randomized double-blind studies have evaluated the efficacy of Gasserian stimulation. A 2001 meta-analysis by Holsheimer [182], which included 267 patients who underwent surgery, found a significant benefit in 50% of patients. The author concluded that the stimulation ‘test’ was a good predictor of long-term success. In 83% of those who tested positive, the decrease in pain was at least 50%; in 70%, the long-term benefit was greater than 75%. Success in those with postherpetic neuralgia was very low (<10%), and improvement appeared to be inversely proportional to sensory loss.

The second modality is suboccipital or C1–C3 stimulation at the root level for the treatment of some types of headache [183], within the spectrum of chronic headaches cited in the International Classification of Headache Disorders (ICHD-H) by the International Headache Society [184,185], for its possible direct action on the trigeminal tract [186].

#### 4.4.4. Chronic, Analgesic Spinal Cord Stimulation (SCS)

Based on the theories proposed by Melzack and Wall [10], Shealey et al. [47] performed the first electrode implantations directly on the posterior spinal cord. Subsequently, due to arachnoiditis, CSF fistulas, and fixation difficulties, the fixation site was moved to the subdural [187] and epidural spaces [188]. All implantation techniques required laminectomy.

Thanks to technological advances, the quality and size of the electrodes were improved, allowing epidural placement using a percutaneous technique, which was originally described by Dooley [189] and rapidly extended (Hosobuchi et al., [190]; Ray [191]; Sedan and Lazorthes [192]).

##### Mechanisms of Action of SCS

The following physiological bases of the SCS are currently accepted as most valid:Antidromic stimulation of the posterior spinal cord and the theory proposed by Melzack and Wall [74], directly on the inhibitory neurons of the SGR or indirectly through the inhibition of T cells in lamina V.Conduction block of the spinothalamic tracts [193].Orthodromic firing of the VPL thalamic nucleus and subsequent negative feedback on reticular thalamic nuclei [49].Raising the level of endorphins, by direct stimulation of the dorsal horns or by activation of higher centers rich in endorphins. These centers, e.g., PAG, can utilize a descending inhibitory pathway, such as the reticulospinal tract [58] and other serotonergic systems [194].

In this sense, administering 5-L hydroxytryptophan or maintaining a diet rich in tryptophan favors a longer duration of the beneficial effect and lower frequency of tolerance phenomena in SCS, as occurs in PAG stimulation [195].

##### Indications and Results

It is extremely difficult to draw a clear conclusion about the indications for SCS and its long-term outcomes. Valuable experience was acquired in the first phase of implementation [188,192,196,197]. There are also more recent references [198,199], to which we must add the recommendations of the European Federation of Neurological Societies (EFNS) [200]. According to the above, a positive result (after a correct selection of the patients), in the long term (longer than two years), is achievable in at least 40–50% of patients affected by the following:Chronic postoperative low back pain (“failed back syndrome” or “low back pain syndrome”), one of the most common causes of disability in middle-aged people.Postamputation pain and phantom limb pain, especially in the lower extremities.Other neuritic pain (post-traumatic, diabetes…).Peripheral deafferentation pain (postherpetic neuralgia, post-cordotomy pain, and brachial plexus avulsion, fundamentally). In these cases, the statistics are few and contradictory.Ischemic pain, which deserves mentioning. SCS in ischemic pain was first proposed by Dooley [189], and later systematized and disseminated by Meglio et al. [201]. SCS not only produced a very significant decrease in pain, but also increased blood flow in the extremities. The explanation for this increase is not yet clear, with several mechanisms proposed [202,203]: (a) antidromic activation of C fibers of the dorsal spinal roots; (b) activation of ascending pathways to higher autonomic centers; and (c) segmental inhibition of sympathetic vasoconstrictor fibers.

Cook et al. [204] reported that SCS effects were longer lasting and more beneficial than those for sympathectomy, without causing irreversible injury.

Murphy and Giles [205], in 1987, proposed SCS for unstable angina pectoris. This indication is reserved for patients with unstable angina that does not respond to other therapeutic measures. Treatment should be carried out under the supervision of a multidisciplinary team that includes cardiology, cardiovascular, anesthesia, and neurosurgery specialists [206,207,208] (Figure 15). DeJongste and Foreman [209] thoroughly described the indications and therapeutic management.

f.Malignant pain. The success rate is very low and recent pharmacological advances raise doubts on the indication for SCS in these patients.g.Other indications. If the type of pain is not included in those for which this modality has greater chances of success, the following conditions described by Long et al. [210] must be met: (a) clear physiological cause of pain; (b) no personality disorders; (c) anxiety and depression, if present, are adequately treated; and (d) previous positive pain control test using TENS.

#### 4.4.5. Analgesic Deep Brain Stimulation (DBS)

The analgesic effect of electrical stimulation of subcortical brain structures was first reported by Heath [211] in 1954 and by Pool et al. [212] in 1956. Electrodes were applied in the septal area, anterior and lateral to the anterior columns of the fornix; the analgesic effect persisted even several days after a single stimulation session. Another suitable location chosen for analgesic stimulation was the caudate nucleus [213]. This structure was later relegated, as the same beneficial and long-term outcomes were not obtained [214].

The first DBS, in the thalamus, was performed by Mazars et al. [215] in 1962. At that time, recording electrodes were used to locate structures for lesion surgery. After noticing that stimulation of the medial lemniscal fibers suppressed pain and VPL nucleus stimulation produced an analgesic effect, these electrodes were left in place for variable periods of up to two months. Adams et al. [48,163] observed the same effect during the localized stimulation of the VPL nucleus; therefore, thalamic electrode implantations began in 1971 [216].

At this time, the locations chosen for analgesic DBS were (1) subcortical somatosensory areas, i.e., lemniscus medialis [217]; sensory thalamic nuclei (VPL and VPM); and the posterior limb of the internal capsule [48,218,219], which were fundamentally indicated for deafferentation pain; and (2) PAG and periventricular gray matter (PVG) [30], generally indicated for pain from ongoing nociception.

##### Thalamic Nuclei

The first electrode implantation for analgesic DBS (Mazars et al. [160,219] was based on previous interventions, in which the stimulation of the fibers of the lemniscus medialis suppressed postherpetic pain. On the contrary, this pain was exacerbated by stimulation of the spinothalamic fasciculus. This analgesic effect was seen to be most effective with stimulation of the ventral posterolateral thalamic nucleus (VPL) (Figure 16).

During the first ten years of application, Mazars et al. [220] left electrodes in this location for a variable period of time, from a few days to a maximum of two months, and noted that (a) the disappearance of pain was complete, lasting several hours or even days after a single stimulation session and beginning one to three minutes after the end of the session; pain recurrence was prevented by performing two or three daily stimulation sessions; (b) the efficacy of stimulation was not exhausted in 20% of cases; and (c) the response was best for individuals with deafferentation pain without large areas of sensory involvement.

In 1966, Adams et al. [221] observed, during the localized stimulation of the VPL nucleus in which they were going to perform a lesion, that the patient not only reported the expected paresthesias, but also pain suppression. After the technique was consolidated, thalamic electrode implantations began to be performed in 1971 [48,163,222]. Likewise, Mazars et al. [220,223] implanted chronic electrodes along with a pulse generator.

##### Implantation Technique

Implantation techniques are widely referred to in the works by Hosobuchi [224,225,226,227]. The target varies with different authors and with the exact location of the painful area [163,192,195,218,228].

##### Indications and Outcomes

The main indication is pain with a pathophysiological mechanism that involves a sensory-discriminative deficit, in which interventions interrupting the nociceptive pathway are contraindicated due to increased deafferentation. Importantly, the larger the area with altered sensory input, the lower the chances of success of thalamic stimulation.

At first, good results were reported by Mazars [160,219,220], Adams [229], and Hosobuchi [225]. They reported success rates ranging from 70% to 80% [192].

The study with the best results and the largest European series of patients was conducted by Mazars et al. [223]. Excluding pain from ongoing nociception, in which stimulation of the VPL and VPM nuclei generated no results, and reviewing more than a hundred implantations in deafferentation pain, the authors distinguished two groups:Deafferentation pain with an extensive area of skin anesthesia. Positive long-term results were achieved in 50% of cases, including lesions of the brachial plexus, postherpetic neuralgia, postrhizotomy or postcordotomy pain, and thalamic syndrome.Other deafferentation pain. This includes peripheral nerve injuries (from wounds, crushing, surgery, radiotherapy, etc.), spinal cord or mesencephalic lesions, either vascular or in multiple sclerosis (resulting in hyperpathy-hyperesthesia syndrome), and traumatic paraplegia.

Other authors also reported optimistic results, although not in such a high percentage, for different deafferentation syndromes [195,218,230,231]. However, it is difficult, by studying the different series, to obtain an accurate conclusion because of the large number of variables involved: electrode placement location, type of pain, follow-up time, scarce number of patients, way of assessing outcomes, etc. The personal experience reported by Adams [232] shows a positive long-term result (mean 5-year follow-up) in 39% of cases of thalamic syndrome, postcordotomy dysesthesia, anesthesia dolorosa, pain in paraplegia, and pain due to peripheral neuropathy. The latter type of pain includes postoperative lumbar arachnoiditis (failed back syndrome), and was treated either with thalamic stimulation alone [233] or combined with PAG stimulation [227].

In summary, stimulation of the sensory thalamic nuclei remains indicated for neuropathic pain due to deafferentation, with a long-term positive outcome in 30 to 50% of patients. Thalamic syndrome pain is only likely to be reduced if the lesion is small. The selection of patients with failed back syndrome for thalamic stimulation alone or combined with PAG stimulation remains controversial.

##### Posterior Limb of the Internal Capsule

This zone was first described by Adams et al. [48] after an accidental observation in the operating room, obtaining an excellent result in his first case of pain due to cortical injury. His theory was to reinforce the cortical inhibitory mechanism, altered in the patient by an injury at this level. This could be best achieved by acting at the level of the internal capsule, which is a higher level than the sensory thalamic nuclei.

As for other locations, at first the results were extraordinary, even for thalamic pain [224]. However, in the long term, efficacy declined [163,225].

The results reported by Adams and Hosobuchi [234] seem to agree that stimulation at this level could be indicated for deafferentation pain: (a) postcordotomy dysesthesias (50% or greater success rate); (b) thalamic pain (30% success rate); and (c) lesions of the cerebral hemispheres causing pain, the results of which are difficult to assess due to their rarity.

##### Mesencephalic Lemniscus Medialis

This technique was primarily performed by Mundinguer [219] and Hosobuchi [225] (Figure 13 and Figure 14). Its purpose was blockading both the spinothalamic and medial lemniscal systems as well as the spinoreticular pathways and afferent system to the pulvinar nucleus. It also seems that its action is mediated by the release of endorphins.

Although it is difficult to evaluate the results due to the small number of patients and short follow-up time after surgery, this method could be indicated for facial anesthesia dolorosa.

##### Central Gray Matter

Reynolds [28], in 1969, discovered the phenomenon of analgesia produced by stimulation of the PAG in rats. Since then, numerous reports have revealed the activation of a descending inhibitory system in dorsal horn neurons [29], mediated by the release of endorphins at this level [54].

Richardson and Akil [30] were the first to test and demonstrate that stimulation of this area in humans produces analgesia. However, they found that it produced nonbeneficial side effects (mainly ocular alterations); therefore, they changed the location of the electrode to the PVG, in the posteromedial part of the thalamus, at the level of the parafascicular nucleus. The analgesic action obtained was bilateral, although with a greater effect on the side contralateral to the electrode. Unlike thalamic stimulation, no paresthesias appeared with stimulation. However, sometimes, a sensation of heat or cold was generated. Long-term results [235] were very good, i.e., reduced pain by more than 50% in 70% of patients with pain from ongoing nociception (mainly malignant pain and postoperative low back pain), and in 40% of patients with deafferentation pain.

Almost simultaneously, Adams and Hosobuchi [234] used PAG and PVG stimulation in a large number of patients. Several conclusions can be obtained from their results: (a) the analgesia produced is reminiscent of that obtained by opioids [225]; (b) analgesia is reversed by the administration of naloxone [33,162]; (c) stimulation induces an increase in endorphins in CSF [225]; (d) there is cross-tolerance with opioids [32]; and (e) analgesia is mediated by a descending inhibitory system originating at the level of the raphe nuclei, and is serotonergic [58]; therefore, the simultaneous administration of tryptophan (serotonin precursor) delays the possible appearance of tolerance phenomena to stimulation [195].

##### Indications and Results

In order to identify the coordinates of selected points and electrode placement locations, we recommend reading specialized bibliographies [30,192,234].

The best responses are obtained for pain from ongoing nociception, for which an excellent method of selection is a morphine test [227]. If the patient has already developed opioid tolerance, the phenomenon must be reversed to obtain good results after PAG stimulation [225].

A prominent indication is postoperative low back pain (failed back syndrome). In Europe [236] and in the US [225], success rates of up to 80% have been reported. However, the results reported by Adams [232] are more realistic, i.e., long-term favorable results for 48% of patients.

Richardson and Akil [235] reported good results for 40% of patients after PAG and PVG stimulation for deafferentation pain. However, the small number of patients in their series with this type of pain, as well as the unsatisfactory results reported by other authors [227], mean that PAG or PVG stimulation for this type of pain is not advised.

Concerning malignant pain, although the results of different series are not bad (about 50% of patients with good results), the indication is questioned because patients are less motivated to continue stimulation as their general condition deteriorates [236]. In addition, with new pharmacological advances and as alternative anesthetic and neurosurgical techniques have become available, the number of patients with this type of pain, who would theoretically benefit from PAG and PVG stimulation, has progressively decreased in successive statistics [237].

In incapacitating postoperative low back pain, the combined stimulation of PAG/PVG and sensory thalamic nuclei has been proposed, with excellent results in more than 50% of cases [226,237].

##### Hypothalamus

Cluster headaches are trigeminal-vascular headaches, at the level of the 1st branch, along with vegetative signs such as conjunctival redness, lacrimation, eyelid edema, and rhinorrhea, lasting several minutes. Sometimes, they are very intense and do not respond to medical treatment, presenting a very dramatic clinical situation. One of the solutions indicated, apart from trying treatments such as radiofrequency of the Gasserian and sphenopalatine ganglions, is hypothalamic stimulation.

This approach was proposed by Leone et al. in 2001 [238,239,240,241]. The discovery of metabolic alterations in this region during attacks led to the implantation of electrodes in one patient, in whom the attacks died out. Subsequently, several series and a randomized double-blind study have been published, with interesting results, showing a decrease of at least 50% in weekly attacks [242].

Other series support these results [243,244,245].

Stereotactic localization can be performed for indications described by these authors and Seijo et al. [246].

##### Reflections and Current State of Analgesic DBS

Similar to other therapeutic approaches in medicine, neurostimulation in its various aspects was born after observing a beneficial, fortuitous event. Then, its more systematic application, the evaluation of favorable effects and of errors, and detailed studies of the provoked phenomena have resulted in a number of physiopathological theories, technological foundations, and indications for application, which are still far from consolidated.

The path which was traveled up until the end of the 1980s has been well studied. New therapeutic knowledge was accompanied by euphoria and a drive to produce short-term and long-term effects. Using a critical attitude and caution with regard to indications and applications, new and very useful treatment modalities were built and offered to many patients suffering from unbearable pain to whom, until then, medicine could not offer anything except comfort and recommendations of patience.

However, several phenomena occurred that almost stopped the application of these techniques. There has been continuous progress in medical treatment in pain clinics, with less aggressive and complex techniques. Additionally, the possible abuse of indications by surgical groups, together with multiple series of small numbers of patients, was led by the ambition to improve their therapeutic capacity, without thinking about other, less plausible forces of professional competition and the harsh rules of subsistence in the economic market.

Thus, relevant published studies have several issues: insufficient sample sizes, limited and nonconsensual use of follow-up scales, inability to locate lesions using neuroimaging techniques, difficulty in blinding and having control groups, excessively short follow-up periods, and incomplete characterization of the patients included in the study and the adverse effects they presented [87].

In fact, analgesic DBS was not approved by the American FDA, and only following its acceptance for the treatment of abnormal movements in 1997 was this therapeutic possibility considered again [247].

In this second stage, starting in the 1990s, several facts should be taken into account:As in the first period, there was a rich basic pathophysiological contribution, simultaneous with published clinical results. In this second stage, there were no great advances in the basic pathophysiological studies advocating analgesic DBS, although reports on clinical advances, based on neuroimaging, have supported what has already been stated.

Thus, in PET and functional MRI studies, PAG-PVG stimulation has been shown to activate the medial thalamus and anterior cingulate. In addition, stimulation of the sensory thalamic nuclei activates the amygdala, insula, and anterior cingulate, along with the primary and secondary sensory cortex [248].

2.Regarding the implantation of intracranial electrodes, there have been great advances that have modified the surgical placement techniques and improved intra- and postoperative control. We refer to the use of CT and MRI fusion programs [249] and new neuronavigation software, including the different stereotactic atlases [250]. In addition, improvements in intraoperative neurophysiological exploration, through prior recording with microelectrodes, compared to classic stimulation with implantable macroelectrodes, should also be mentioned [251]. Microelectrode recording and stimulation are very useful at the level of the sensory thalamic nuclei, in which there is a somatotopic distribution; however, it is not as useful as at the PAG-PVG level [250].3.DBS was restricted during the two preceding decades to a few centers in Europe and America. Despite this, in the last decade of the last century, more than 1000 patients were treated using this procedure [252].

The clearest indications with the highest success rates are brachial plexus injury, phantom limb pain, peripheral neuropathy, and failed back syndrome [250,253]. The positive effect declines over time [249].

No results were obtained for spinal cord lesions [254]—obtaining better outcomes with SCS [252]—or for central thalamic pain [247], for which Katayana et al. [255] proposed stimulation of the posterior oval nucleus.

Regarding the pain type/target relationship, nociceptive pain responds best to PAG-PVG stimulation (>50% of patients), and neuropathic pain responds best to sensory thalamic nuclei stimulation [247]. Combining stimulation in both targets increases the efficacy for some types of pain, such as failed back syndrome [256]. The results for facial deafferentation pain are highly variable, depending on the surgical group [247].

#### 4.4.6. Analgesic Motor Cortex Stimulation (MCS)

As seen, several pain entities, all within neuropathic pain, do not respond well to DBS: central post-thalamic infarct pain, facial neuralgia with a deafferentation component, and spinal cord injuries.

Aiming to improve these options, in 1991, Tsubokawa et al. [257] proposed MCS as an alternative to the treatment of thalamic pain. This approach was endorsed in a new publication two years later. The positive response was confirmed by other authors, who also extended the indication to neuropathic facial pain [258,259] (Figure 17).

This new therapeutic possibility was based on the following reasoning [260]: 1—Central deafferentation pain is a type of neuropathic pain involving a complete or incomplete lesion of the spinothalamic tract, without lesion of the posterior cord-medial lemniscus pathway. It appears several months after the lesion, being distributed in the same area of sensory loss, and does not respond to morphine—only, in very few cases, to the therapies already mentioned. 2—Hyperactivity in the neurons of the sensory pathway above the lesion has been found in animals [261,262] and in humans [263]. 3—This hyperactivity can be reduced by MCS [262]. Volkers et al. [264] reviewed the literature and found that after switching the iMCS electrode ‘ON,’ increased rCBF occurred in the (1) anterior cingulate gyrus; (2) putamen; (3) cerebral peduncle; (4) precentral gyrus; (5) superior frontal gyrus; (6) red nucleus; (7) internal part of the globus pallidus; (8) ventral lateral nucleus of the thalamus; (9) medial frontal gyrus; (10) inferior frontal gyrus; and (11) claustrum, as compared to the “OFF” situation.

The idea was to implement this therapeutic approach by the epidural placement of electrodes on the primary motor area (Brodmann Area 4) and proceed with chronic stimulation, similar to CASCS or ADBS. De Ridder et al. [265] also proposed stimulation of the primary somatosensory cortex.

##### Indications and Outcomes

This modality is indicated in patients with neuropathic pain, preferably thalamic, in the trigeminal area, peripheral area, or after spinal cord injury. It has the usual conditions of duration of clinical pain: no response to conventional therapies, psychological normality, acceptance of the new therapy without generating false expectations and, if possible, adequate response to morphine (no decrease in pain) and barbiturates (decrease in pain) tests [260].

Again, the experience presented in the different series varies, although there are commonalities:From the point of view of scientific evidence, the series have better methodological quality than the DBS series [87]. For this procedure, double-blind tests were carried out because patients do not notice anything during stimulation, allowing the treatment to be switched on and off [266,267,268], finding a placebo effect in 35% of patients [269].Although it is difficult to summarize all the series, the best responses (up to 70% of patients with more than a 50% reduction in pain intensity) occurred in the following order: facial neuralgia, brachial plexus avulsion, peripheral neuropathies, spinal cord injury, thalamic pain, and phantom limb pain) [266,270,271].Efficacy decreases over time [272,273], requiring a good knowledge of the stimulation possibilities [274]. For example, for thalamic lesions, the response was better if there was no complete loss of strength in the contralateral limbs [260].This technique does not reduce pain to zero for all patients. However, like DBS, it has a very low complication rate, with greater than 60% of patients reporting excellent or good results regarding their pain (reduction of more than 50%, with improved quality of life) [273].The mechanism of action involves activating the inhibitory system and modulating the activity of the anterior area of the corpus callosum (pgACC, pregenual anterior cingulate cortex) and the PAG [264].

A growing interest is emerging about the changes that chronic pain could produce at the level of the cerebral cortex. Novel advances in neuroimaging techniques demonstrated structural, functional, and neurochemical changes [275].

This could be an important step in the search for new approaches, new targets, and new technologies to treat chronic pain diseases, in addition to having more objective control of the results of current neurosurgical treatments.

## 5. Final Comment

From a neurosurgical perspective, an important question remains: Do we leave patients with intractable pain without an alternative, after not responding to all the different therapeutic possibilities offered today by modern pain clinics?

These are the patients with the most difficult and frustrating type of pain to treat. They come to us after years of suffering from pain, which has been modified by multiple treatments. They have even suffered iatrogenic complications that have made it worse. They carry complex behavioral problems. Could neurostimulation (SCS, DBS or MCS) be the last resort? [276].

Neurosurgeons should consider offering these neurostimulation modalities, which have a very low level of serious complications and real possibilities of benefiting at least 50% of a well-selected population of patients.

## Figures and Tables

**Figure 1 brainsci-12-01584-f001:**
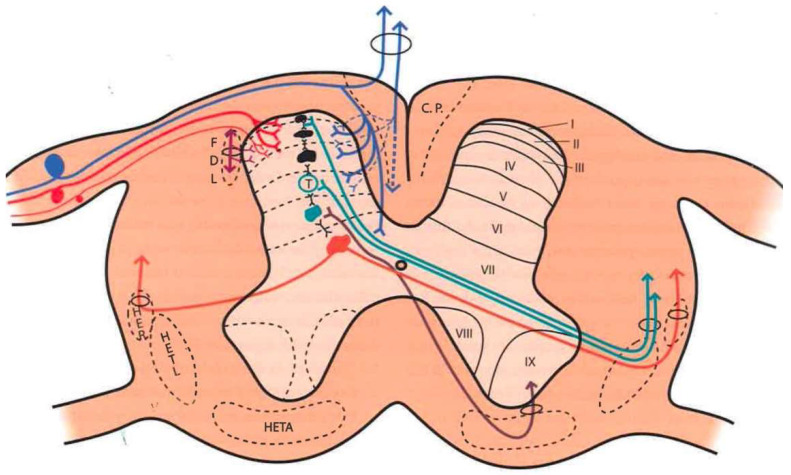
Distribution of afferent pathways in the spinal cord (see text). (Taken from [16]). C.P. = P.C. (Posterior cord). HER = SRT (Spinoreticular tract). HETL = LSTT (Lateral spinothalamic tract). HETA = VSTT (Ventral spinothalamic tract). FDL = DLF (Dorsolateral fasciculus, Lissauer’s tract). I-IX = Laminae of spinal cord gray matter.

**Figure 2 brainsci-12-01584-f002:**
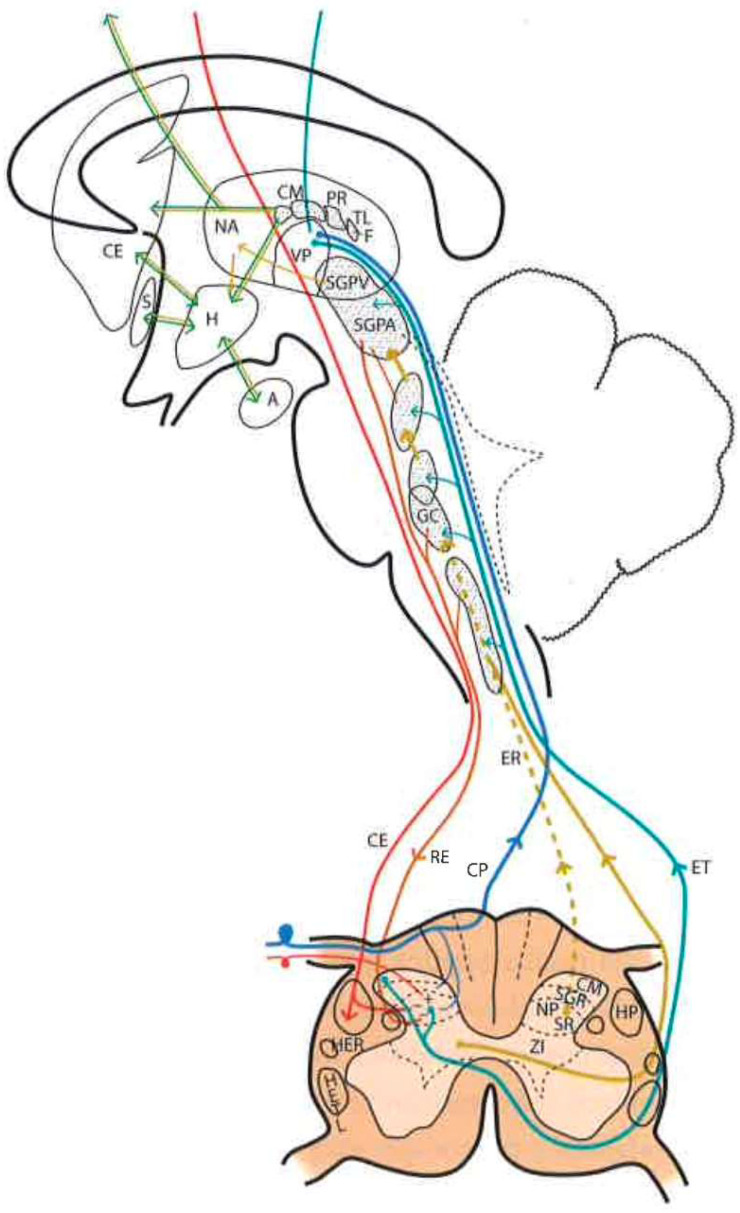
Diagram of the distribution of the main sensory pathways involved in pain perception and control (see text). (Taken from [16]). SGPV = PVG (Periventricular Gray Matter). SGPA = PAG (Periaqueductal Gray Matter). H = H (Hypothalamus). CE = CS (Corticospinal), RE = RS (Reticulospinal), ER = SR (Spinoreticular) and HETL = LSTT (Lateral spinothalamic) Tracts. NP = PN (Pulvinar nucleus). GM = MG (Medial geniculate nucleus).

**Figure 3 brainsci-12-01584-f003:**
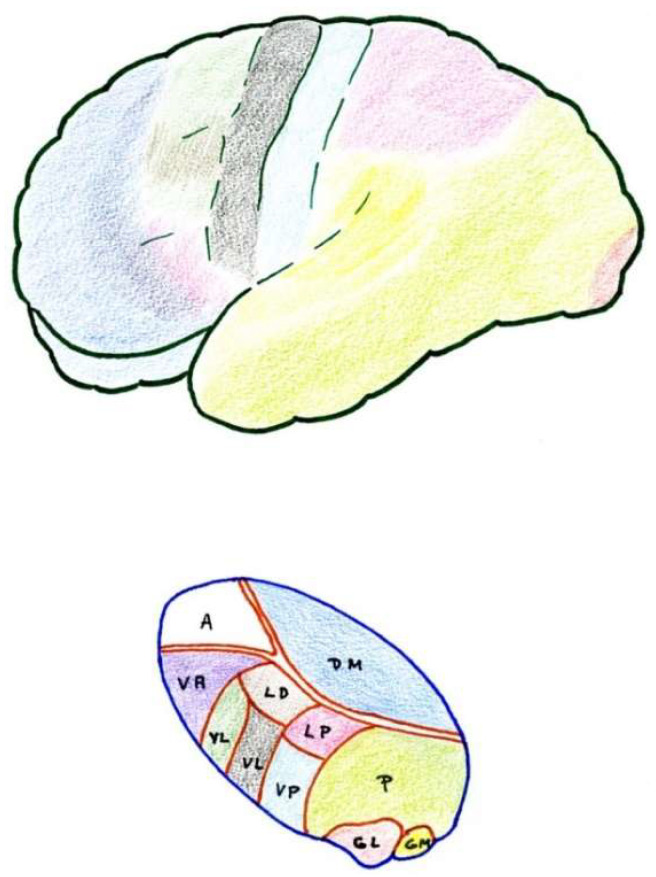
Anatomical–functional correlation between the thalamus and cerebral cortex (see text). LD (Lateral dorsal nucleus). LP (Lateral posterior nucleus). VL (Ventral lateral nucleus). VA (Ventral anterior nucleus). VL = VPL (Ventral posterolateral). VP = VPM (Ventral posteromedial). A = AN (Anterior nuclear group). DM = MD (Medial dorsal nucleus). P (Pulvinar). GL = LG (Lateral geniculate nucleus). GM = MG (Medial geniculate nucleus).

**Figure 4 brainsci-12-01584-f004:**
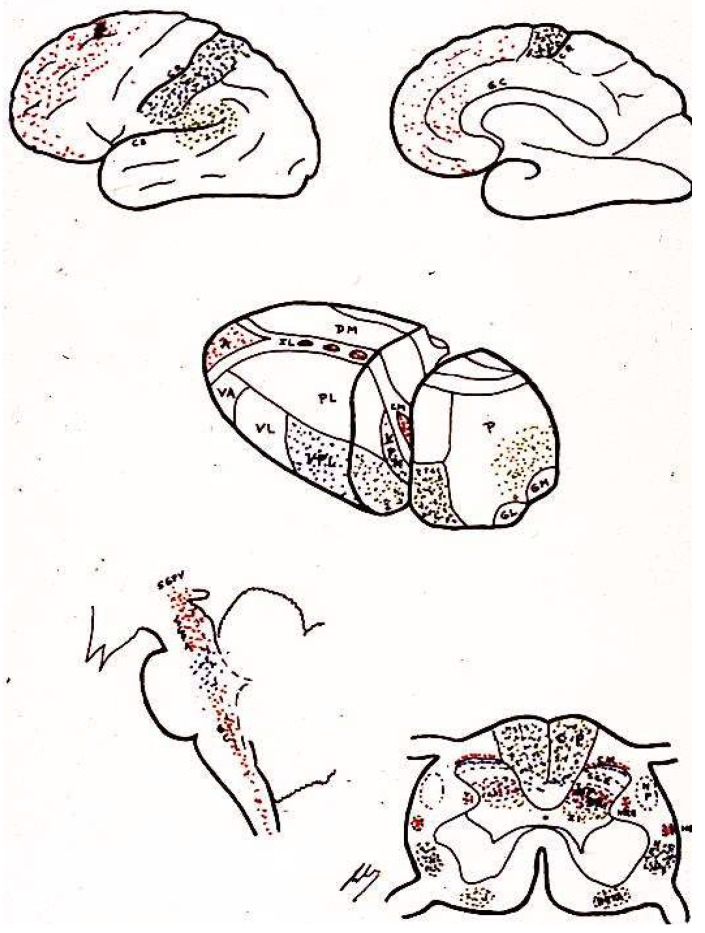
Area distribution according to the neurotransmitters involved (see text). LD (Lateral dorsal nucleus). LP (Lateral posterior nucleus). VL (Ventral lateral nucleus). VA (Ventral anterior nucleus). VL = VPL (Ventral posterolateral). VP = VPM (Ventral posteromedial). A = AN (Anterior nuclear group). DM = MD (Medial dorsal nucleus). P (Pulvinar). GL = LG (Lateral geniculate nucleus).

**Figure 5 brainsci-12-01584-f005:**
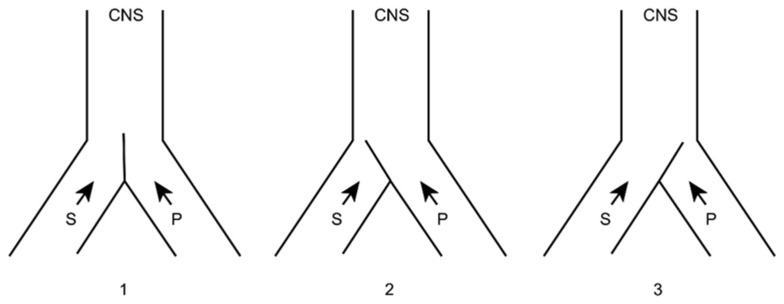
Melzack and Wall gate control theory. 1: Normal. 2: Increased pain afferents. 3: Increased sensory afferents. (see text). CNS: Central Nervous System. S: Sensory afferents. P: Pain afferents.

**Figure 6 brainsci-12-01584-f006:**
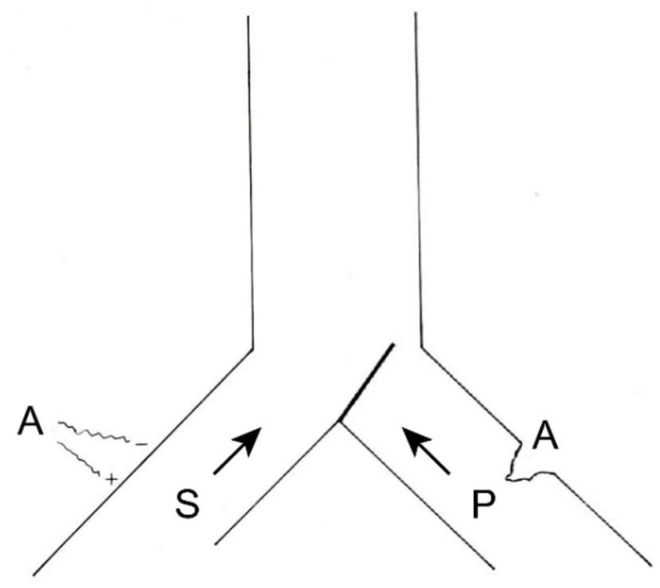
Therapeutic possibilities for reducing pain afferents. On the right, surgical interruption of pain pathways; on the left, increasing sensory afferents through neurostimulation. (see text).

**Figure 7 brainsci-12-01584-f007:**
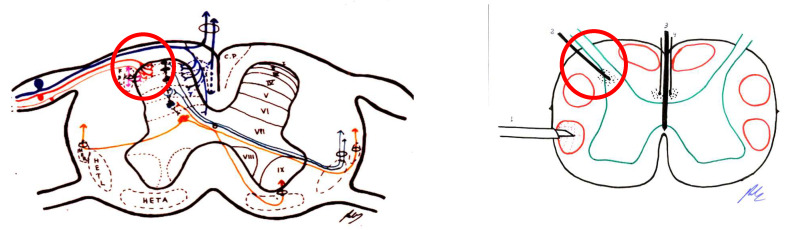
DREZ to be altered (see text). C.P. = P.C. (Posterior cord). HER = SRT (Spinoreticular tract). HETL = LSTT (Lateral spinothalamic tract). HETA = VSTT (Ventral spinothalamic tract). FDL = DLF (Dorsolateral fasciculus, Lissauer’s tract).

**Figure 8 brainsci-12-01584-f008:**
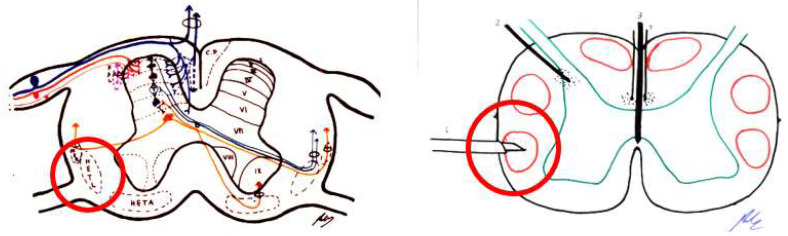
Lesion zone in the spinothalamic tract (see text). C.P. = P.C. (Posterior cord). HER = SRT (Spinoreticular tract). HETL = LSTT (Lateral spinothalamic tract). HETA = VSTT (Ventral spinothalamic tract). FDL = DLF (Dorsolateral fasciculus, Lissauer’s tract).

**Figure 9 brainsci-12-01584-f009:**
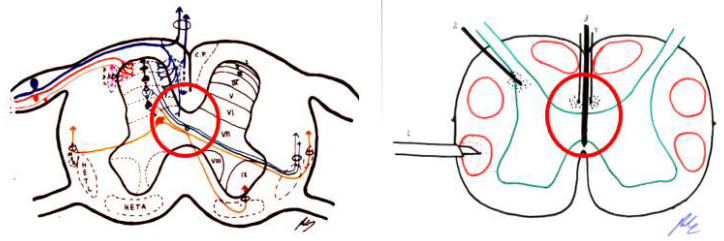
Myelotomy (see text). C.P. = P.C. (Posterior cord). HER = SRT (Spinoreticular tract). HETL = LSTT (Lateral spinothalamic tract). HETA = VSTT (Ventral spinothalamic tract). FDL = DLF (Dorsolateral fasciculus, Lissauer’s tract).

**Figure 10 brainsci-12-01584-f010:**
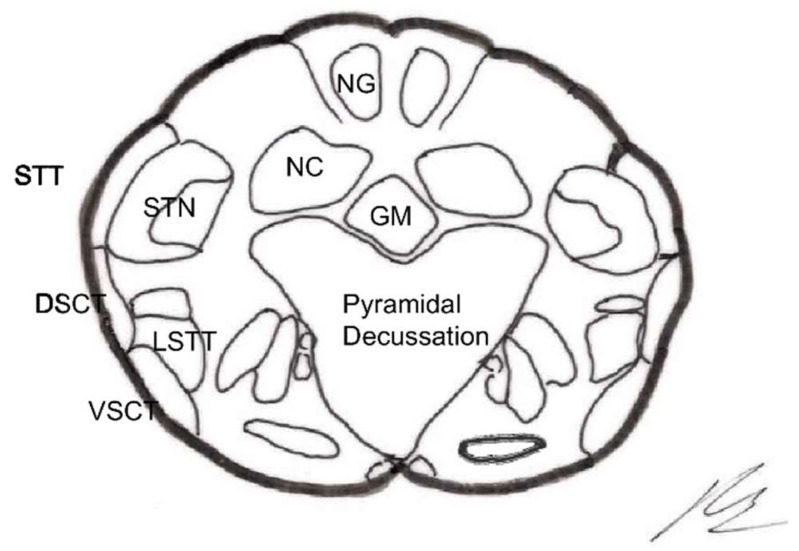
Section of the trigeminal spinal tract (TST) at the level approximately 11 mm below the obex (modified from [122]). GN—Gracile Nucleus. CN—Cuneate Nucleus. STrigT—Spinal Trigeminal Tract. STrigN—Spinal Trigeminal Nucleus. AECT—Anterior Spinocerebellar Tract. PECT—Posterior Spinocerebellar Tract. ATLS—Anterolateral System. CG—Central Gray.

**Figure 11 brainsci-12-01584-f011:**
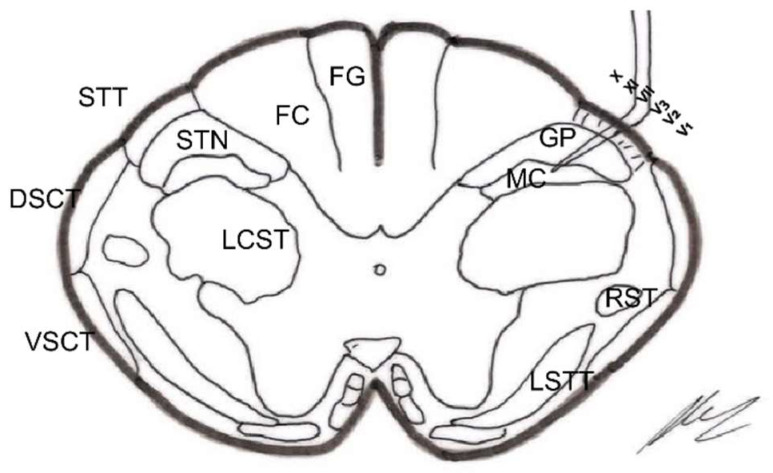
Trigeminal nucleotomy (see text). Section approximately 18 mm below the obex (modified from [122]). GF—Gracile Fasciculus. CF—Cuneate Fasciculus. STrigT—Spinal Trigeminal Tract. STrigN—Spinal Trigeminal Nucleus. AECT—Anterior Spinocerebellar Tract. PECT—Posterior Spinocerebellar Tract. LCET—Lateral Corticospinal Tract. GP—Gelatinosa Portion of Spinal Trigeminal Nucleus. MC—Magnocelllular Portion of Spinal Trigeminal Nucleus. RST—Rubrospinal Tract. ALS—Anterolateral System.

**Figure 12 brainsci-12-01584-f012:**
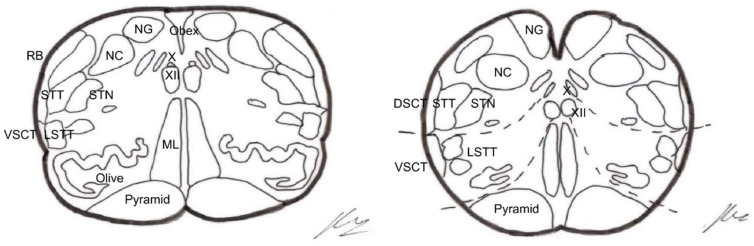
Sections at the level of the obex and 2–3 mm below (modified from [122]). GN—Gracile Nucleus. CN—Cuneate Nucleus. RB—Restiform Body. STrigT—Spinal Trigeminal Tract. STrigN—Spinal Trigeminal Nucleus. AECT—Anterior Spinocerebellar Tract. ALS—Anterolateral System. ON—Olivary Nucleus. ML—Medial Lemniscus. P—Pyramid. X—Dorsal Motor Nucleus of Vagus. XII—Hypoglossal Nucleus.

**Figure 13 brainsci-12-01584-f013:**
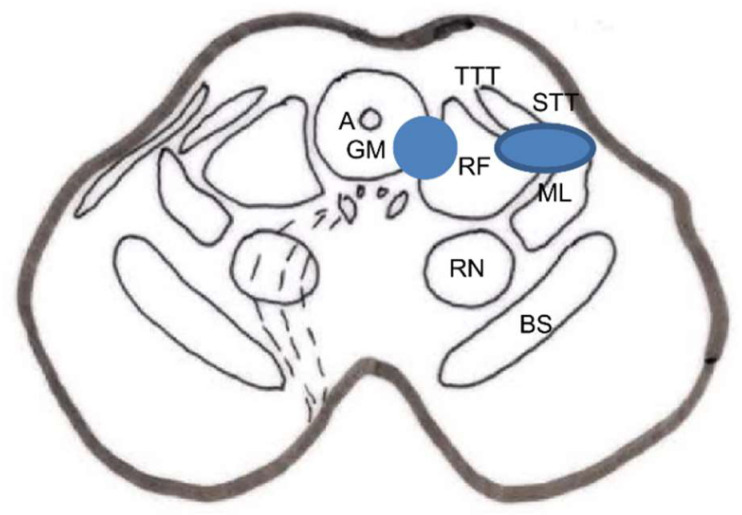
Two types of lesions proposed for stereotactic mesencephalotomy: medial or extralemniscal [128] and lateral [127] (spinothalamic tract) (Diagram taken from [129]). A—Aqueduct. CG—Central Gray. ML—Medial Lemniscus. RN—Red Nucleus. SN—Substantia Nigra. STT—Spinothalamic Tract. STrigT—Spinal Trigeminal Tract.

**Figure 14 brainsci-12-01584-f014:**
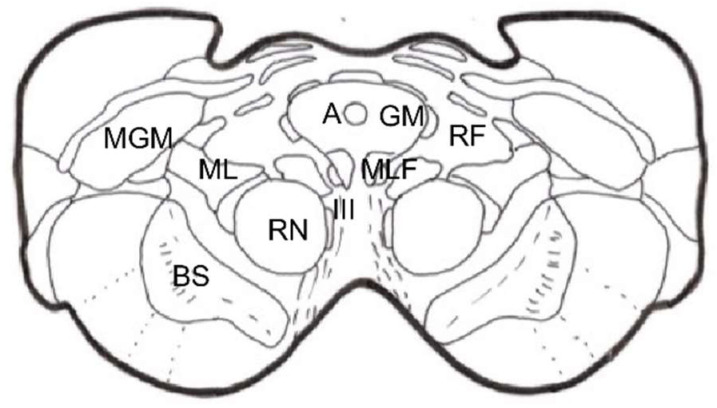
Mesencephalic section at the level proposed for mesencephalotomy. (Modified from [122]). The actual anatomical complexity of the areas represented in the diagram in Figure 13 is shown. A—Aqueduct. CG—Central Gray. ML—Medial Lemniscus. MLF—Medial Longitudinal Fasciculus. MGM—Medial Geniculate Nucleus. RF—Reticular Formation. RN—Red Nucleus. SN—Substantia Nigra.

**Figure 15 brainsci-12-01584-f015:**
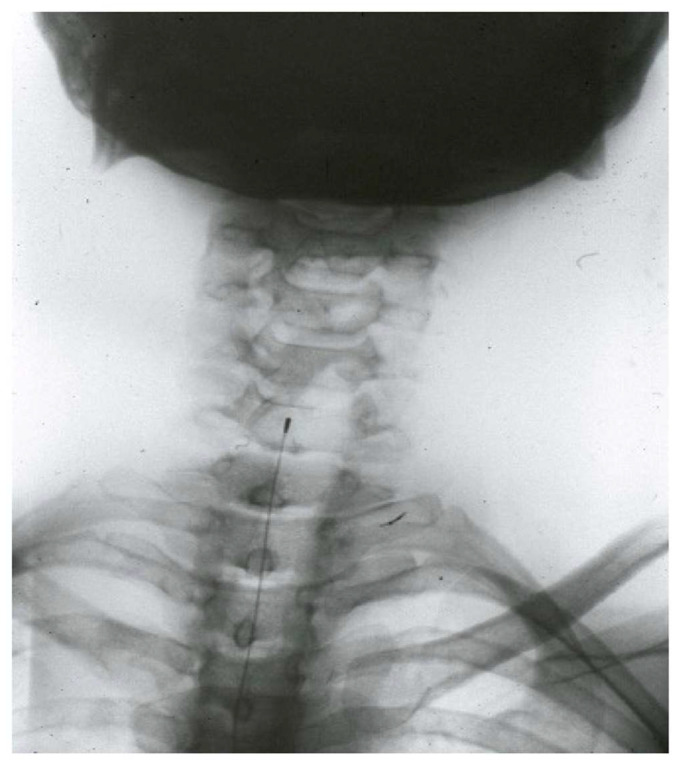
Cervical stimulation in a patient with unstable angina.

**Figure 16 brainsci-12-01584-f016:**
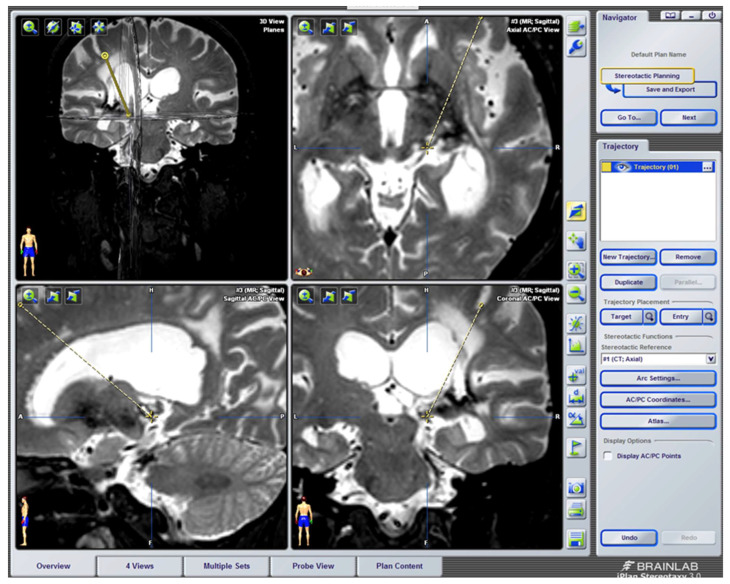
DBS at the level of the VPL nucleus in a female patient with Dejerine syndrome, with more disabling pain in the right lower extremity, on the sole of the foot. It prevented her from walking. She has reported good results for more than 5 years. Within the coordinates of the VPL nucleus, an area without hypointensity was located in T2, obtaining a sensory response to stimulation in the pain area.

**Figure 17 brainsci-12-01584-f017:**
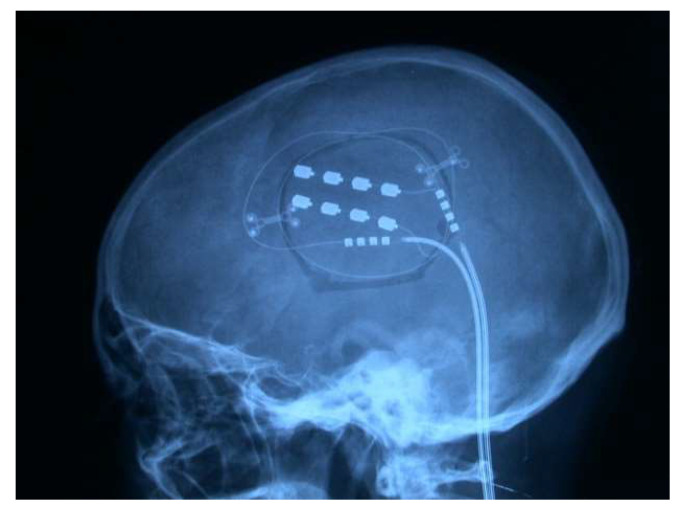
Control after placing epidural electrodes over the motor cortex.

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
