# Peer review of "Neurosurgical Treatment of Pain"

_brainsci, 2022, doi:10.3390/brainsci12111584_

Round 1

Reviewer 1 Report

I had the pleasure to review the manuscript submitted by Sola and Pulido entitled "Neurosurgical Treatment of Pain".

Although based on a classic topic, this manuscript has the advantage of reviewing the neurosurgical approaches to pain. I have found particularly interesting the physiology part as well as the description of the available surgical procedures.

I have found the paper well written and easy to read. Drawings, images and tables are appropriate. References are adequate.

In general I have no specific suggestion and in my opinion this paper deserve to be published as it is.

Author Response

Thanks a lot for your comments.

Reviewer 2 Report

I have read this manuscript with great interest, and I  have to ascertain that article is very very good, topic was presented in exceptional way. References (278 cited articles) were selected properly. The form of the presentation will make this review  very attractive for readers. Authors very thoroughly analysed surgical treatment of pain. All chapters of the article were methodologicly prepared, and presented. In my opinion tme manuscript :"Neurosurgical Treatment of Pain", is worth to be publish in Brain Sciences.

Author Response

I would like to thank you for your comments.

Sincerely yours

Reviewer 3 Report

I would like to congratulate the authors for their work. I believe this wide overview about pain would help clinicians and professors (in fact, I would recommend it to my students for sure).

Although I understand is hard to discuss in detail most of the concepts showed due to the extension but, as a suggestion, I would appreciate a more detailed explanation regarding the cortical changes associated with chronic pain and the mechanisms behind the electrical neuromodulation (also percutaneous since is widely used in physical therapy). Maybe this lectures can assist you in these issues "https://www.mdpi.com/2379-139X/8/5/180; https://www.mdpi.com/2077-0383/11/13/3753"

Author Response

I appreciate your comments.

I have carefully read the works referred to.

It seem to me that one of these works can very well complete our review

So, I refer it to the end, with a vew to the future.
